# Optimizing IoT Intrusion Detection Using Balanced Class Distribution, Feature Selection, and Ensemble Machine Learning Techniques

**DOI:** 10.3390/s24134293

**Published:** 2024-07-01

**Authors:** Muhammad Bisri Musthafa, Samsul Huda, Yuta Kodera, Md. Arshad Ali, Shunsuke Araki, Jedidah Mwaura, Yasuyuki Nogami

**Affiliations:** 1Graduate School of Environmental, Life, Natural Science and Technology, Okayama University, Okayama 700-8530, Japan; 2Green Innovation Center, Okayama University, Okayama 700-8530, Japan; 3Faculty of CSE, Hajee Mohammad Danesh Science and Technology University, Dinajpur 5200, Bangladesh; 4Graduate School of Computer Science and Systems Engineering, Kyushu Institute of Technology, Fukuoka 804-8550, Japan

**Keywords:** intrusion detection system, feature selection, class balancing, ensemble technique, stacked long short-term memory

## Abstract

Internet of Things (IoT) devices are leading to advancements in innovation, efficiency, and sustainability across various industries. However, as the number of connected IoT devices increases, the risk of intrusion becomes a major concern in IoT security. To prevent intrusions, it is crucial to implement intrusion detection systems (IDSs) that can detect and prevent such attacks. IDSs are a critical component of cybersecurity infrastructure. They are designed to detect and respond to malicious activities within a network or system. Traditional IDS methods rely on predefined signatures or rules to identify known threats, but these techniques may struggle to detect novel or sophisticated attacks. The implementation of IDSs with machine learning (ML) and deep learning (DL) techniques has been proposed to improve IDSs’ ability to detect attacks. This will enhance overall cybersecurity posture and resilience. However, ML and DL techniques face several issues that may impact the models’ performance and effectiveness, such as overfitting and the effects of unimportant features on finding meaningful patterns. To ensure better performance and reliability of machine learning models in IDSs when dealing with new and unseen threats, the models need to be optimized. This can be done by addressing overfitting and implementing feature selection. In this paper, we propose a scheme to optimize IoT intrusion detection by using class balancing and feature selection for preprocessing. We evaluated the experiment on the UNSW-NB15 dataset and the NSL-KD dataset by implementing two different ensemble models: one using a support vector machine (SVM) with bagging and another using long short-term memory (LSTM) with stacking. The results of the performance and the confusion matrix show that the LSTM stacking with analysis of variance (ANOVA) feature selection model is a superior model for classifying network attacks. It has remarkable accuracies of 96.92% and 99.77% and overfitting values of 0.33% and 0.04% on the two datasets, respectively. The model’s ROC is also shaped with a sharp bend, with AUC values of 0.9665 and 0.9971 for the UNSW-NB15 dataset and the NSL-KD dataset, respectively.

## 1. Introduction

The Internet of Things (IoT) includes physical objects with sensors and software that connects and shares data with other devices through the internet. These features enable them to collect, transmit, and receive data. Typically, these data are utilized for interaction with and control and observation of the real environment. Data collected through these devices can be analyzed locally or sent to the cloud via gateways or edge devices [1]. IoT devices facilitate communication, data sharing, and automated actions across various domains, including homes, industries, cities, healthcare, agriculture, transportation, and retail, leading to their extensive deployment [2].

This growth has also led to more traffic in cyberspace and a rise in advanced intrusion attacks. IoT system attacks can lead to significant problems that impact both the targeted devices and the broader network infrastructure and compromise data integrity and privacy and even pose risks to physical safety. These attacks exploit vulnerabilities within IoT systems, making it easy to launch various cyber threats such as DDoS attacks, botnets, malware infections, and ransomware [3]. It is essential to safeguard the IoT infrastructure against potential threats to minimize the risks of intrusion attacks on IoT systems. This can be accomplished by implementing intrusion detection systems (IDSs). An IDS is like a digital watchdog for networks. It carefully watches for any unusual activity and alerts administrators if it finds anything suspicious. Moreover, an advanced IDS can spot both known and new types of threats, making it essential for keeping networks safe.

A conventional IDS operates using signature and anomaly detection methods [4]. These traditional approaches have several inherent limitations. Firstly, the rules or signatures used for detection require frequent updates to match the continuously changing environment of cyber threats. Failure to update these rules promptly can lead to missed detection of new or modified attacks. Second, these systems’ low accuracy frequently results in a high rate of false positives, which mistakenly classify benign activities as threats while failing to detect actual threats. Thirdly, conventional IDSs tend to generate a high number of false alarms. This can make security analysts feel exhausted from dealing with alerts, causing them to miss real threats among all the false ones. To address these challenges, implementing IDSs with machine learning (ML) and deep learning (DL) techniques has been proposed [5,6,7,8,9,10,11].

ML and DL techniques have the potential to significantly improve the performance of IDSs. By training these models on large datasets of network traffic data, they are able to recognize intricate patterns and irregularities that could point to malevolent behavior [12]. Unlike traditional rules-based systems, ML and DL models can adapt and evolve as new threats emerge, providing a more robust and proactive approach to threat detection. One key advantage of using ML and DL for IDSs is their ability to handle high-dimensional and heterogeneous data sources. IoT systems generate vast amounts of data from various devices and sensors, making it challenging for traditional methods to effectively analyze and correlate this information. ML and DL models can process and extract meaningful insights from these diverse data sources, enabling more comprehensive and accurate detection of potential threats across the entire IoT infrastructure.

However, several challenges may arise that can impact the performance and effectiveness of these models. First, when a model overfits to training data and is unable to generalize well to new, unknown data, it performs poorly in terms of identifying real threats or producing an excessive number of false positives [13]. Second, the presence of unimportant or irrelevant features in network traffic data can introduce noise and obscure meaningful patterns [14]. Third, large datasets with many features, which are common in IoT and network environments, can lead to higher computational costs and longer training times for ML/DL models, which can be particularly challenging in resource-constrained IoT devices or edge computing environments with limited computational power, storage, and memory [15]. Addressing these challenges is crucial for the successful implementation of ML/DL techniques for IDSs in IoT systems and requires careful model selection, tuning, and optimization to ensure optimal performance, accuracy, and efficiency while considering the constraints of the target environment.

In this study, we employ the UNSW-NB15 dataset and the NSL-KDD dataset and perform essential data preprocessing steps to prepare the data for analysis. Firstly, we address the class imbalance issue by balancing the class distribution to prevent bias towards the majority class in the model. This step is crucial for accurate anomaly detection in IoT networks, where malicious activities may be underrepresented in the data. Secondly, we employ feature selection techniques to identify and retain the most relevant features from the dataset. This not only improves the model’s performance by reducing noise and irrelevant information but also reduces computational costs and training times, which is particularly important in resource-constrained IoT environments.

To optimize the construction of an effective intrusion detection system (IDS) for IoT networks and tackle the challenges mentioned above, we train two ensemble models: one using a support vector machine (SVM) with bagging and another using long short-term memory (LSTM) with stacking. The SVM model is created by combining multiple SVM classifiers, each trained on different subsets of the data using bagging. The LSTM model is created by combining multiple LSTM models using stacking. This model can handle sequential data, learn complex features, generalize well, and integrate effectively with ensemble methods [16]. The models are evaluated with several parameters, including accuracy, precision, recall, F-measure, overfitting value, and ROC curve. We also monitor the computation time during training, as real-time anomaly detection is crucial for IoT networks. Based on these metrics, the LSTM stacking model with ANOVA selection proves to be the superior model, demonstrating the most accurate, reliable, and efficient anomaly detection capabilities. Additionally, we implement the model on a Raspberry Pi 3 Model B+ and measure the model’s loading time.

Our paper presents the following contributions:We use a class balancing approach to address biased models that perform poorly on minority classes.We use a feature selection approach to improve the prediction performance and reduce complexity, resulting in faster training times and reduced computational resources.We evaluate the proposed approach on ML and DL IDSs designed for binary classification: considering that the dataset consists of numerous features, the goal is to identify only features that are highly correlated with the class.We evaluate the performance of the SVM bagging and LSTM stacking models using several parameters, including accuracy, precision, recall, F-measure, overfitting value, ROC curve, model size, and computation time during training. We also measure the loading time on a Raspberry Pi 3 Model B+.

This article is divided into several sections. Section 2 provides a preliminary introduction, including a dataset and related work. The methodology is presented in Section 3, followed by the experiments and discussion in Section 4. Finally, the conclusion is presented in Section 5.

## 2. Preliminaries

In this section, we present the description, features, and exploration of the datasets. The following section presents a literature review and the methodologies and novelty of the related works.

### 2.1. Datasets

Many simulated datasets have been developed during the last few decades to tackle a variety of issues. The majority of these datasets replicate the key characteristics of actual network traffic [17]. Some widely used datasets for assessing IDS performance are DARPA98, NSL-KDD, ADFA, CIC-IDS2017, and UNSW-NB15. Based on trends and the availability of a number of threats relevant to IoT networks, this paper evaluates the proposed method using the UNSW-NB15 and NSL-KDD datasets in light of IoT network threats. The primary goal of using these two datasets is to evaluate the DL-IDS method.

#### 2.1.1. UNSW-NB15 Dataset

The UNSW-NB15 dataset [18,19] is used for training and evaluating the proposed framework. This dataset includes both normal and abnormal IoT device traffic. The abnormal traffic can help with detecting potential attacks on IoT devices and networks. Data are provided in various formats: pcap, Argus, Bro, and CSV files [19]. A pcap file is used as the primary source for analyzing data characteristics on the network. It is extracted into CSV files using Zeek IDS and Argus [18]. Each record is classified as either normal or attack, and it has 45 features. The nine categories of attack types are as follows: Fuzzers, Analysis, Backdoors, DoS, Exploits, Generic, Reconnaissance, Shellcode, and Worms [18]. These attack types can be relevant to IoT security, especially concerning network-based attacks against IoT devices. This dataset is also commonly used for analyzing packets, especially for IDS systems [3,8,11,18,19,20,21,22,23,24,25].

It has 257,673 rows: the number of rows for each category is shown in Table 1. The types of attacks listed below could be relevant to IoT security, especially when it comes to network-based attacks against IoT devices. The UNSW-NB15 dataset captures network traffic data in a lab environment and includes traffic patterns and anomalies that may be relevant to IoT environments. Therefore, the dataset is useful for analyzing network-based attacks targeting IoT devices. It is appropriate for training and evaluating anomaly detection and IDS, which are critical components of IoT security.

#### 2.1.2. NSL-KDD Dataset

Researchers have extensively utilized the NSL-KDD dataset to address the IDS problem [14,24,25]. This dataset contains 41 features and includes separate testing and training sets. Both sets contain randomized data of both categorical and numerical nature. Notably, the probability distributions of the training and test sets differ, providing a closer simulation of real-world application conditions. The characteristics are categorized as either normal or attack, with specific indication of the attack type, and attacks can be classified into three primary categories: basic, content-based, and traffic-based [26]. The attack categories fall into four main groups: Denial of Service (DoS), User to Root (U2R), Remote to Local (R2L), and Probe. The NSL-KDD dataset’s test set does not contain any duplicated records, which enhances the learning process for algorithms and contributes to better detection rates. With 148,517 rows, the detailed breakdown of each category is shown in Table 2.

### 2.2. Related Works

Most IDS research has been evaluated only with known datasets and has tended to ignore unknown data. Recently, new IDSs based on ML and DL have been proposed as a solution to overcome this limitation. These IDSs use ensemble learning techniques to improve their performance and accuracy in detecting unknown data. Several studies have demonstrated that ensemble methods provide more precise and accurate results compared to a single model.

XGBoost has been used on the UNSW-NB15 dataset to determine feature importance and create an optimal feature vector. Additionally, several supervised machine learning models for IDS have been applied. A binary classification has been used to evaluate the performance of each machine learning algorithm. An SVM method with a radial basis function (RBF) kernel achieved 75.42% accuracy with only 13 features, while Kasongo et al. achieved 70.98% accuracy using optimal features [23]. The XGBoost feature selection technique can enhance model performance. However, the model exhibits poor precision, with a value of 58.89%. This issue occurs due to the imbalance in class proportions, leading to poor performance in detecting the minority class.

Convolutional neural networks (CNNs) are utilized for the analysis of spatial and hierarchical characteristics within a dataset. Bidirectional long short-term memory (Bi-LSTM) layers are employed to investigate the long-term temporal attributes of the data. By combining these two techniques, it becomes possible to predict potential attacks. Evaluation of the binary classification results of the proposed model for the UNSW-NB15 dataset is conducted using stratified k-fold cross validation, with variations in k values from 2 to 10. According to the study by Sinha et al. [24], the outcomes include a false positive rate of 7.70%, an accuracy of 93.84%, and an average detection rate of 94.70%. The deep learning model put forth shows promise for improving the performance of IDSs. Nevertheless, it is crucial to recognize the challenges and limitations associated with the model, such as training complexity, interpretability, and resource requirements.

An oversampling technique for intrusion detection using GAN and feature selection was proposed to address data imbalances and high dimensionality of datasets. Gradient penalty Wasserstein GAN (WGAN-GP) generated attack samples, and a subset of features was selected based on analysis of variance. ANOVA can improve intrusion detection model accuracy by eliminating redundant and irrelevant features from the dataset. The proposed model was evaluated using the NSL-KDD, UNSW-NB15, and CICIDS-2017 datasets. Accuracy and F-measure are used as metrics to evaluate the detection performance of machine learning models. Liu et al.’s results showed that an ML model’s detection performance can be improved through WGAN-GP and ANOVA [25]. However, the evaluation metrics did not consider data training and testing, so the paper cannot demonstrate the performance on both datasets.

The goal of an ensemble learning strategy is to combine the advantages of feature selection and individual detection algorithms to improve the performance of IDSs. The CSE-CIC-IDS2018 dataset is used to compare the univariate chi-square test and Spearman’s rank correlation coefficient in order to increase a system’s accuracy and decrease its detection time. It was discovered that detecting highly associated features is more successfully accomplished via Spearman’s rank correlation coefficient. A decision tree and logistic regression were the basic classifiers that were employed. The Spearman’s rank correlation coefficient was used to choose a dataset with 23 features. Fitni et al.’s model reduced the detection time from the initial 34 min and 2 s to 10 min and 54 s while achieving an accuracy of 98.8% [5]. However, the evaluation metrics did not consider the receiver operating characteristic (ROC). A more thorough assessment of the model’s performance is provided by this evaluation metric, which also shows the model’s performance across all feasible thresholds.

The synthetic minority oversampling technique (SMOTE) is utilized to compare the performance of balanced and imbalanced datasets across multiple classifiers. In imbalanced datasets, standard classifiers focus on the majority class and ignore the minority class. This issue has been addressed by researchers who have proposed several solutions, with most of their efforts focusing on binary-class problems. One study aimed to find the best sampling rate for oversampling in imbalanced datasets with different minority class samples. The F1 measure is a better performance metric for imbalanced datasets as it considers both recall and precision measures, unlike accuracy. The SMOTE technique is used to overcome this issue and to enhance the precision of the dataset. Among all the algorithms tested, the SVM algorithm performed the best in F1, recall, and AUC evaluations. This is due to the SVM algorithm’s ability to manage a substantial number of characteristics in a dataset that Alfrhan et al. designated as CICIDS2017 [27]. However, there is no information about model performance on training and testing datasets, so we cannot determine how well the model generalizes to unseen data.

## 3. Proposed Methodology

The proposed IDS includes preprocessing, feature selection, classification methods, and evaluation. Firstly, the dataset is preprocessed by encoding categorical data to numerical values, normalizing it to the same scale, and balancing the dataset using SMOTE. Additionally, to maintain the high performance of the IDS while reducing classification overhead, we use feature selection techniques to select the most important features. We use two feature selection techniques: namely, Spearman rank correlation and ANOVA. We use the UNSW-NB15 and NSL-KDD datasets for evaluating the proposed model. Both datasets comprises many features, some of which have little or no impact on intrusion identification.

The proposed intrusion detection system (IDS) utilizes two ensemble models: one using SVM with bagging and another using LSTM with stacking. Finally, we use the confusion matrix to assess the model’s performance and determine which model is superior to the others. We also evaluate the model in terms of model size and time for loading the model on a Raspberry Pi 3 Model B+. Figure 1 shows the proposed framework, and the subsection that follows explains our process.

### 3.1. Preprocessing

In the machine learning pipeline, preprocessing is an essential step. The data must be prepared and transformed. Four steps were involved: data encoding, normalization, class balancing, and feature selection.

#### 3.1.1. Data Encoding

The DL and ML algorithms only work with numerical values, so features with categorical values must be transformed into numerical data. Categorical features are converted to integers with values between 0 and S−1. *S* represents the number of symbols. Table 3 shows the numerical values of categorical data that have high-cardinality categorical features. To achieve this, we utilized label encoding to prevent an increase in the number of features, as a larger number of features could impact computational complexity [28]. Label encoding helps streamline training by avoiding feature explosion with one-hot encoding, thus ensuring that the dataset size and computational requirements remain manageable.

#### 3.1.2. Data Normalization

Both datasets have attribute values that cover a wide range. This can cause errors and have a detrimental effect on the model’s performance. To tackle this problem, standardization and normalization are two methods that can be used for scaling the features. In our investigation, the min–max scaling method applies a linear transformation to the original data, which helps to develop a model. The basic formula $x^{\prime }\;=\;\frac{x-min(x)}{max(x)-min(x)}$ can be used to discover the minimum and maximum values within a range of [0, 1], where *x* represents the original value and *x′* represents the normalized value. Table 4 shows the normalized data for the dur and sbytes features for the first 10 data points of the UNSW-NB15 dataset.

#### 3.1.3. Class Balancing

Models with imbalanced datasets may outperform with respect to the majority class while neglecting or misclassifying the minority class [29]. Balancing the dataset helps to ensure that the model learns to recognize and predict both classes accurately: this has been proven by increasing the F1 score and recall metric [27,30]. Imbalanced learning is addressed using resampling techniques such as oversampling, undersampling, combined oversampling and undersampling, and ensemble sampling [31].

To address the imbalance in class distribution, SMOTE creates new data points for the minority class by interpolating feature values between the minority sample and its nearest within-class neighbors. The SMOTE is often used as a benchmark for oversampling [27,31,32]. It creates synthetic data points for the minority class by generating new data points; these are not simply duplicates but are synthetic data points. This helps to prevent overfitting and is an improvement over simple random oversampling [30]. The detailed distribution of the balanced data is explained in Section 4.

#### 3.1.4. Feature Selection

The process of feature selection involves selecting relevant features while removing irrelevant ones from the original dataset. This eliminates redundant information and reduces computational cost [33,34]. Accurate detection performance depends greatly on feature selection, which is an effective technique for reducing the impact of irrelevant variables and noise [35]. The proposed IDS classification utilizes a feature selection algorithm to identify significant features that have a strong impact on the classes. In this paper, Spearman rank correlation and ANOVA are used; these analyze the strengths of relationships between variables.

The statistical measure known as Spearman’s rank correlation coefficient is used to ascertain whether two variables have a monotonic connection [27]. This measure helps with predicting one variable based on another. Feature selection is done using correlation, as highly correlated variables are good predictors of the target variable. Spearman’s rank correlation values range from −1 to 1. A high distance rank score indicates a strong positive correlation and importance of the feature. A threshold was established to determine which features should be included in the model after each feature was given a score based on statistical evaluations [5]. This approach is suitable for data with different scales of measurement, as shown in Table 4, as it reduces the impact of extreme values and discrepancies in measurement scales on the correlation analysis.
(1)$$\rho =\frac{\sum_{i}\left(x_{i}-\bar{x}\right)\left(y_{i}-\bar{y}\right)}{\sqrt{\sum_{i}{\left(x_{i}-\bar{x}\right)}^{2}\sum_{i}{\left(y_{i}-\bar{y}\right)}^{2}}}$$

Equation (Equation 1) represents the correlation coefficient (ρ) between two vectors *X* and *Y*, where $x_{i}\left(1,2,\ldots ,n\right)$ and $y_{i}\left(1,2,\ldots ,n\right)$ are the samples for the random variables X and Y, respectively. If the correlation coefficient (ρ) is close to ±1, it indicates strong association between the two features. In this case, one of the features can be retained. On the other hand, if the value of ρ is close to 0, it signifies that there is no association between the two features, and both features should be filtered out [19,36].

ANOVA is a statistical method used to compare the means of independent groups [22]. This method ranks the features by calculating the ratio of variances within groups and between groups [6]. The one-way ANOVA F-test is a statistical tool used to identify significant differences between the means of two or more groups [7,36], which can help with the classification of traits. ANOVA is a suitable method for selecting features in the network log that contribute to distinguishing between normal and attack instances in network traffic data simultaneously [20]. It effectively leverages the dataset’s characteristics for feature selection, allowing the identification of significant features for distinguishing between normal and attack instances.

The procedure for feature selection uses a dataset (*D*) that contains *n* rows. Each row in the dataset has *k* continuous values for categorical variables. For an individual *j* belonging to the group *i*, $y_{ij}$ denotes the value, and ${\bar{y}}_{j}$ denotes the mean. The term $\bar{y}$ is the representation of the mean of the entire dataset, and $n_{i}$ denotes the total number of values in a group. The F-test compares two types of variances: the mean sum of squares between groups (MSB) and the mean sum of squares within groups (MSW) [22,37]. Equation (Equation 2) shows the F-test. $S_{E}$ refers to the sum of squares within groups, and it can be expressed as (Equation 3). $S_{A}$ refers to the sum of squares between groups. It is a statistical measure that is used to evaluate the variability between group means, and it can be expressed as (Equation 4) [35]. The ANOVA *F-value* is calculated for each feature and class variable, and we select *K* features with the strongest connections to the class using the *F-value*.
(2)$$F=\;\frac{S_{E}\left(D\right)/\left(k-1\right)}{S_{A}\left(D\right)/\left(n-k\right)}$$
(3)$$S_{E}\left(D\right)=\;\sum_{i=1}^{k}\sum_{j=1}^{n_{i}}{{(y}_{ij}-{\bar{y}}_{j})}^{2}$$
(4)$$S_{A}\left(D\right)=\;\sum_{i=1}^{k}n_{i}{{(\bar{y}}_{i}-\bar{y})}^{2}$$

The detailed features of each feature selection method are explained in Section 4.

### 3.2. Classification Using Ensemble Techniques

Ensemble techniques combine multiple methods for training the dataset. In this research, we utilize two types of ensemble: namely, SVM with bagging and LSTM with stacking.

#### 3.2.1. SVM with Bagging

Support vector machine (SVM) is a popular algorithm used for binary classification [6,8,9]. In the field of IDS, transactions are classified as either normal or intrusions, irrespective of the type of attack. This study utilizes SVM due to its advantages in analyzing high-dimensional spaces. In addition, SVMs only use a portion of the decision function’s training points: they are memory-efficient.

To separate data points of different classes, SVM finds the optimal hyperplane in a high-dimensional feature space [38]. The maximum margin hyperplane is selected to maintain the maximum separation from the closest data points for every class. Four different types of kernels are used in SVM: sigmoid, polynomial, radial basis function (RBF), and linear [39]. In this study, we used the RBF-SVM algorithm, which is a powerful and versatile machine learning algorithm that offers flexibility, robustness, and strong generalization performance.

A single SVM model may not always learn the exact parameters for the global optimum [40]. It is possible that not all unknown test samples can be correctly classified using the support vectors acquired during the learning process. Therefore, a single SVM model may not provide optimal classification for all test examples.

To address the limitations of SVMs, in this work, we adopt a bagging technique to create an ensemble of diverse samples using bootstrapping sampling [38]. To establish the final predicted class in bagging, many SVMs are trained individually using bootstrap techniques, and then they are aggregated via majority voting. Training set $TR\;=\;\left\{\left(x,y\right)|i=1,2,3\ldots l\right\}$ for a single SVM consists of pairs of data points *x* and their labels *y*, where *l* is the total number of datasets. In bagging, an SVM ensemble with *K* independent SVMs is built using *K* training sets of samples. To get a bigger improvement in the aggregation outcome, we must vary the training sample sets. We employ the bootstrap technique to do this. Given a training dataset *D* with *N* samples, we generate *M* bootstrap samples $D_{1},D_{2},\ldots ,D_{M},$ each containing *N* samples drawn with replacements from *D*. In any specific replicate training dataset, an example *x* from the provided training set TR may appear once, more than once, or not at all. A particular SVM will be trained using each replicate training set. The predictions made by each SVM classifier are then combined using majority voting to determine the final predicted class [10]. The overall model for SVM bagging presented in this paper is illustrated in Figure 2.

Each dataset was trained using SVM bagging with 10 base estimators and 10 bootstrap samples. The bagging ensemble consisted of 10 individual SVM models trained on different bootstrap samples of the data. For each test instance, we made predictions using all the SVM models and aggregated the predictions using a majority vote to obtain the final prediction for each instance.

#### 3.2.2. LSTM with Stacking

Recurrent neural networks (RNNs) with long short-term memory (LSTM) are trained to handle the vanishing gradient problem and capture long-term dependencies between data points [37]. The memory cell is a key component of LSTM and is capable of storing information for extended periods. The information flow into and out of LSTM cells is managed by three gates: the input gate, forget gate, and output gate [41]. Figure 3 depicts the LSTM memory cell used in this study.

The input gate ($i_{t}$) regulates how many of each input element enters the cell state. Each input element is passed through a sigmoid activation function, which generates a value between 0 and 1, as represented in Equation (Equation 5). The forget gate ($f_{t}$) plays a crucial role in deciding which information of the cell state is removed from or kept for the model, as represented in Equation (Equation 6). The primary function of this algorithm is to keep track of the previous cell state ($C_{t-1}$) that will be allocated to the current time ($C_{t}$). The term $o_{t}$ is responsible for deciding the amount of the current state that will be passed, as represented in Equation (Equation 7). Initially, the sigmoid layer (σ) defines the output information. Then, tanh processes the cell state and multiplies it by the layer output sigmoid to generate the final output [42].
(5)$$i_{t}\;=\;\sigma \left(W_{ix}x_{t}+W_{ih}h_{t-1}+b_{i}\right),$$
(6)$$f_{t}\;=\;\sigma \left(W_{fx}x_{t}+W_{fh}h_{t-1}+b_{f}\right),$$
(7)$$o_{t}\;=\;\sigma \left(W_{ox}x_{t}+W_{oh}h_{t-1}+b_{o}\right),$$
where the weight is denoted by *W*, the hidden state of the cell at time *t* is represented by $h_{t}$, the input layer is denoted by $x_{t}$, and the bias is represented by *b* [43].

During training, LSTM networks are trained using gradient descent and backpropagation through time (BPTT) algorithms [41,44]. The LSTM cell parameters (weights and biases) are adjusted by propagating gradients through the network to minimize the loss function. Their ability to capture long-term dependencies and effectively handle sequential data makes them well-suited for tasks requiring memory and context preservation over extended sequences.

This helps to build deeper and more sophisticated models that can effectively capture the complex temporal patterns and dependencies present in sequential data. By leveraging hierarchical representations and increased model capacity, LSTM stacking offers a powerful framework for tackling a wide range of sequential learning tasks with improved performance and generalization capabilities.

In this research, we implemented an LSTM stacking network consisting of two LSTM layers connected using the hyperparameter settings illustrated in Figure 4 [11]. The hyperparameters for configuring the LSTM stacking network include the number of hidden layers, dropout rate, activation functions, and dense functions. Each LSTM layer processes the input sequence and passes its output sequence to the next layer in the stack. The two LSTMs have different sizes, with the first one being larger. The first layer focuses on more general features by using 128 hidden layers and 0.3 for the dropout rate, while the second layer targets more specific features by using 32 hidden layers and 0.3 for the dropout rate. The initial LSTM layer processes each dataset. LSTM Layer 1’s output sequence is the input sequence for LSTM Layer 2. LSTM Layer 2’s output sequence is further processed by additional layers: namely, ReLu as an activation function and softmax as a dense layer for binary classification (normal or attack).

To prevent overfitting in our model, we added dropout layers and established feedforward connections in each LSTM layer. We measured the difference between predicted and actual probabilities using sparse categorical crossentropy and combined the LSTM model with an Adam optimizer for improved performance. To prevent overtraining, we stopped training the model when its performance on the validation data ceased to improve by applying early stopping [11].

## 4. Experiment and Discussion

The proposed framework was trained and evaluated on the requirements shown in Table 5. We utilized the Pandas and NumPy libraries for preprocessing tasks and the Matplotlib library to visualize the dataset. Additionally, we utilized the scikit-learn and Keras frameworks for data analysis. To activate the GPU, we used TensorFlow.

### 4.1. Evaluation Criteria

We evaluated the proposed model’s performance using a confusion matrix. The details of these variables are in Table 6. Network flows can be classified as either normal or attack. This problem involves classifying data into two categories, and there are four possible outcomes. A true positive (*TP*) case occurs when the IDS detects an actual attack correctly. A true negative (*TN*) represents an instance when the IDS correctly identifies that no attack is occurring. A false positive (*FP*) case occurs when the IDS incorrectly identifies normal activities as attacks. A false negative (*FN*) case occurs when the IDS fails to detect an actual attack [45].

*Accuracy (Acc):* This metric measures how often the model predicts the correct class for both positive and negative classes. The following formula can be used to obtain it:
$$Accuracy=\;\frac{TP+TN}{TP+TN+FP+FN}.$$*Recall (R):* This metric measures how well the model correctly identifies positive classes. To obtain *recall* values, we can use the following formula:
$$Recall=\;\frac{TP}{TP+FN}.$$*Precision (P):* This metric measures how well a model makes correct predictions for the positive prediction out of the total number of positive predictions. To obtain *precision* values, we can use the following formula:
$$Precision=\;\frac{TP}{TP+FP}.$$*F-measure (F):* The F-measure is a metric that represents the balance between *precision* and *recall*. The following formula can be used to obtain it:
$$F-measure=\;2*\frac{Precision*Recall}{Precision+Recall}.$$*Overfitting:* A model can contain too much information about the training data and not enough about new data, which can lead to poor performance of the model on new data. To obtain overfitting values, we can use the following formula:
Overfitting=trainingaccuracy−validationaccuracy.ROC curve: A binary classifier system’s performance when the threshold for differentiating between positive and negative occurrences is changed can be shown visually via the receiver operating characteristic (ROC). The true positive rate (TPR) against the false positive rate (FPR) relationship is shown on the graph at different threshold settings [19].Model size: It is important to assess the size of a DL/ML model since this can affect memory usage, storage, and energy efficiency. Larger models require more memory, storage, and energy, which can be a problem for devices with limited RAM, like the Raspberry Pi.Loading time: The loading time of a model is the duration it takes for the model to be loaded from storage into memory and be ready for inference. This metric is crucial for applications that need fast startup times, such as real-time systems or edge devices like the Raspberry Pi. The loading time includes the effects of memory caching, which enhances performance for repeated loads. To evaluate the model, we deployed it on a Raspberry Pi 3 Model B+ with the specifications shown in Table 7.

### 4.2. Experimental Results

The proposed approach was balanced using SMOTE and by using two feature selection techniques. The dataset was trained using two ensemble techniques: SVM bagging and LSTM stacking.

#### 4.2.1. Balancing the Datasets

Both datasets are imbalanced: containing different numbers for each class (normal and attack). In this situation, the IDS model may prioritize detecting more frequent traffic over minority attacks, leading to high overall accuracy and yet low detection of minority attacks. This phenomenon is referred to as the accuracy paradox, which highlights that the accuracy value may not reflect the model’s actual performance [21]. To ensure an equitable distribution of data points among different classes, the SMOTE was used. The aim of class balancing is to create a more equitable and representative dataset that leads to better model performance, reduces bias, and improves generalization. The balanced data for normal and attack networks are shown in Table 8 and Table 9.

#### 4.2.2. Feature Selection

In the study, we employed Spearman’s rank correlation and ANOVA techniques to analyze the data. For the Spearman’s rank correlation experiment, we used a threshold of 0.8 to identify strong correlations. ANOVA utilizes the F-test to assess group differences. By using Equation (Equation 2), we set a significance level (α) of 0.05 and select features with *p*-values less than 0.05, as these features are statistically significant and contribute significantly, so they are selected as important features. Table 10 shows the total number of selected features for each dataset.

#### 4.2.3. Evaluation

Two distinct sets were created from the original dataset: the training set, which included 80% of the data, and the validation set, which had the remaining 20%. The UNSW-NB 15 dataset and the NSL-KDD dataset were used to evaluate how well the model performs. The experiment involved creating two models for each dataset. The first model, referred to as “Model 1”, utilized SVM bagging with an RBF kernel and was implemented using the *BaggingClassifier* [8]. This model also compared between no feature selection and feature selection based on the Spearman rank correlation. Table 10 shows the total number of features selected for each dataset. The feature selection method may efficiently choose relevant features for attack classification, which leads to a notable decrease in the total number of features.

Furthermore, there is superior performance across all metrics when implementing feature selection. Moreover, the model size is mostly decreased through feature selection, which is also a positive outcome. In Table 11, Model 1 using the original features of the NSL-KDD dataset showed poor performance. For instance, the model only achieved an accuracy of 53.51%, an F-measure of 34.99%, and a model size of 251,452 Kb. On this dataset, the model also takes 17.1 s to load. However, using the Spearman rank correlation improved the accuracy and loading time significantly, albeit with the highest model training time of 6990.8 s. The Spearman rank correlation also reduces the model’s size. However, when we selected features from the UNSW-NB15 dataset using the Spearman rank correlation, we achieved impressive results, with an accuracy and F-measure of 92.48% and 92.36%, respectively. The model size was 28,095 Kb, and the training time was 596.5 s.

The second model, referred to as “Model 2”, used ANOVA to select important features and utilized LSTM stacking for training. The details of the LSTM architecture are shown in Figure 4. We trained the model for 1000 epochs and stopped early at epoch 51 by using an early stopping technique. We also used this model to compare between the performance without feature selection and the performance using ANOVA for feature selection. The total number of selected features for each dataset is illustrated in Table 10. Our proposed model achieves higher than 95% for all of the classification measures, as Table 12 demonstrates. Using ANOVA for feature selection produced better results, with throughputs of 96.59% and 96.92% for training and validation, respectively, on the UNSW-NB15 dataset and the same performance values for accuracy, recall, and F-measure. Regarding training time, ANOVA is also faster than without feature selection, with training times of 2478.2 s and 2698.4 s respectively. ANOVA also impacts both the model size and the loading time.

On the other hand, the NSL-KDD dataset had the same values for all performance metrics, including accuracy, precision, recall, and F-measure having values higher than 99%. The results show that ANOVA also can increase the performance of the model. The model size was 1278 Kb, and the training time was 2330 s. When implemented on a Raspberry Pi, it takes 4.7 s for ANOVA to load the model.

#### 4.2.4. Discussion

We evaluated overfitting by comparing model performance on the training and validation datasets. Figure 5 shows how well the models perform over the two datasets; Model 2 outperforms Model 1. Model 1 has a huge difference in accuracy between the training and validation datasets. Various parameters in LSTM can be adjusted to improve model performance. Simply increasing the number of hidden layers or units in a neural network does not necessarily result in overfitting. Insufficient layers and units can lead to low training and validation accuracy, while an excess can lead to high training accuracy but low testing accuracy. The ideal balance for a problem lies between these two extremes. In our research, we discovered that the architecture illustrated in Figure 4 provided good performance on both training and validation data. Therefore, it can be concluded that this architecture can generalize the model. On the UNSW-NB15 dataset, Model 1 exhibits an overfitting value of 7.55%, while on the NSL-KDD dataset, it demonstrates an overfitting value of 18.36%. Contrastingly, Model 2 showed overfitting values of 0.33% and 0.04% for the respective datasets. This indicates that Model 2 is highly suitable for detecting anomalies in data that have not been previously seen.

Model 2 has a sharp bend, as depicted in Figure 6. The performance is close to perfect separation, with an AUC (area under the curve) value of 0.9665 on the UNSW-NB15 dataset and 0.9971 on the NSL-KDD dataset. The model successfully distinguishes between positive and negative classes for both the training data and the unknown data, as indicated by the AUC value being close to 1. For the LSTM stacking model, achieving a high AUC suggests that the model has learned complex patterns and dependencies within the data, which makes it highly effective at making predictions. According to our research, achieving high AUC values is achievable through training LSTM stacking models to properly select hyperparameters such as the number of layers, epochs, learning rate, batch size, and hidden units, which can significantly impact model performance. Tuning these hyperparameters effectively achieves better results. In our study, we utilized long epochs along with early stopping, allowing the model to stop training when its performance did not significantly improve. This approach helps us identify the optimal number of epochs required for a good solution.

Regarding the implementation, we evaluated the model in terms of size and loading time. The results showed that Model 1 outperformed Model 2 in terms of loading time on the Raspberry Pi 3 Model B+, and Model 2 outperformed Model 1 in terms of accuracy, precision, recall, F-measure, overfitting, ROC, and model size. The most important aspect of DL-IDS is how well the model can classify traffic. So we firmly believe that the proposed method that uses SMOTE for balancing the dataset, ANOVA for feature selection, and LSTM stacking effectively selected relevant features and reliably classified the attacks in the dataset. In future work, we will optimize the model’s size and loading time.

## 5. Conclusions

In this paper, we propose a scheme to optimize IoT intrusion detection using a combination of class balancing and feature selection for preprocessing. The SMOTE is used to balance the rare classes of the dataset. In addition, we apply Spearman rank correlation and ANOVA to identify the essential features that have a high impact on the class while reducing data dimensionality and computational overhead. We evaluate the performance of SVM bagging and LSTM stacking algorithms on the UNSW-NB15 dataset and NSL-KDD dataset, specifically focusing on accuracy, overfitting, and AUC for binary classification. It is important to note that the training time can impact the model size and overfitting. The performance results suggest that the LSTM stacking with ANOVA feature selection model is superior for classifying network attacks. This model also has a small size and loads quickly, making it suitable for implementation on a Raspberry Pi 3 Model B+.

To enhance our model’s robustness and accuracy, future work will focus on implementing additional deep learning architectures. Specifically, we plan to integrate Transformer models and gated recurrent units (GRUs). For implementation on the Raspberry Pi, the limited computational resources, including CPU, memory, and storage, pose significant challenges. To address these constraints, we intend to optimize the model in order to reduce model size and quickly load the model without significantly compromising performance. This optimization strategy is essential for achieving efficient computation times on the resource-constrained Raspberry Pi platform.

## Figures and Tables

**Figure 1 sensors-24-04293-f001:**
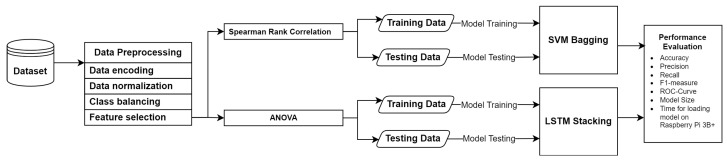
Proposed framework to optimize an IDS.

**Figure 2 sensors-24-04293-f002:**
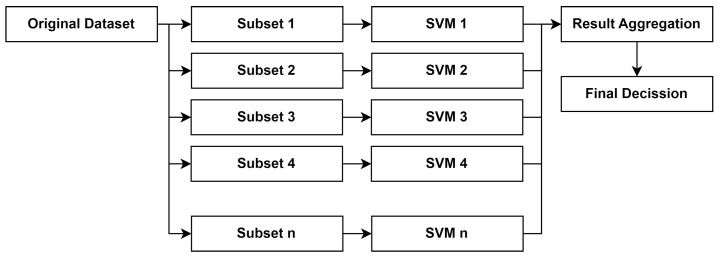
Proposed SVM bagging.

**Figure 3 sensors-24-04293-f003:**
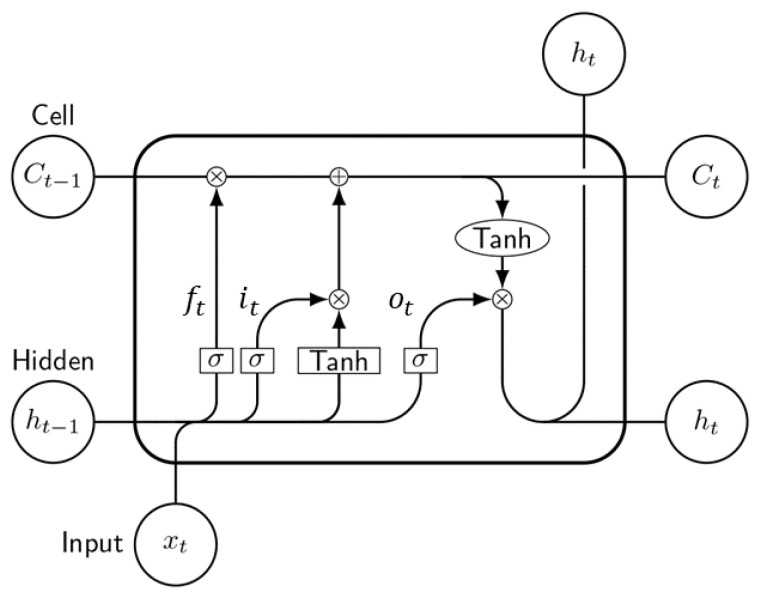
Proposed LSTM network.

**Figure 4 sensors-24-04293-f004:**
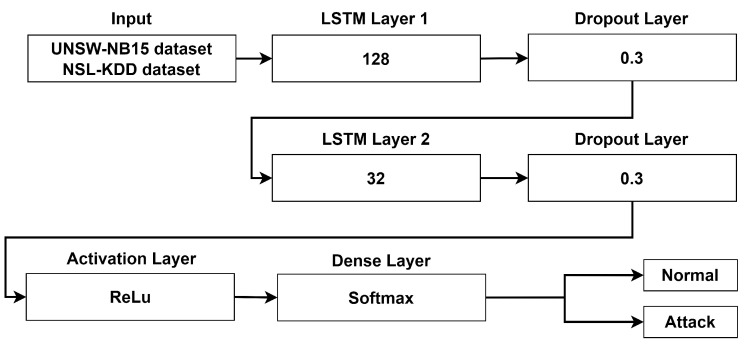
The detailed structure of LSTM stacking.

**Figure 5 sensors-24-04293-f005:**
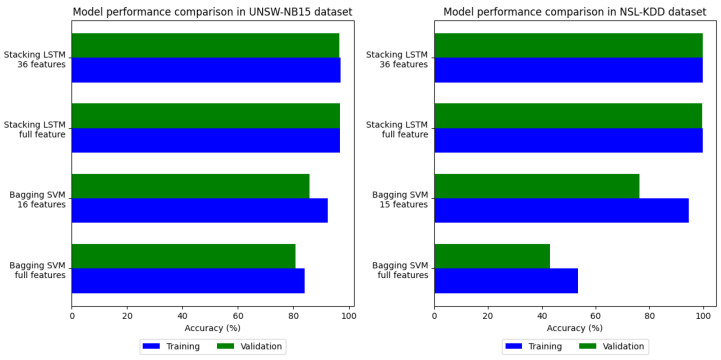
Overall performance of each model.

**Figure 6 sensors-24-04293-f006:**
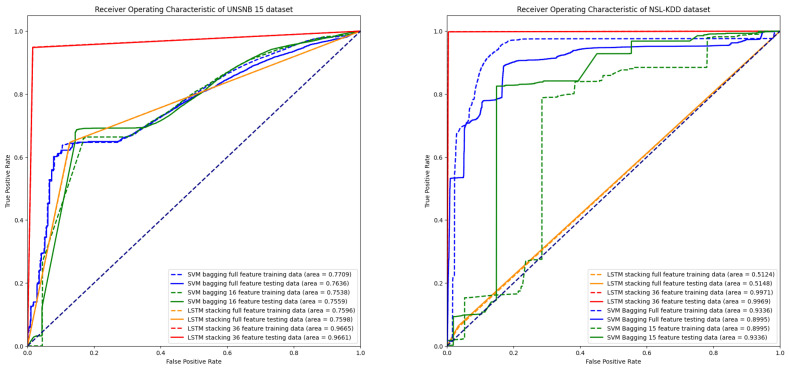
ROC curve comparison.

**Table 1 sensors-24-04293-t001:** Distribution of UNSW-NB15 classes.

No.	Category	UNSW-NB15-Testing-Set	UNSW-NB15-Training-Set
1	Fuzzers	18,184	6062
2	Analysis	2000	677
3	Backdoors	1746	583
4	DoS	12,264	4089
5	Exploits	33,393	11,132
6	Generic	40,000	18,871
7	Reconnaissance	10,491	3496
8	Shellcode	1133	378
9	Worms	130	44
10	Normal	56,000	37,000
Total		175,341	82,332

**Table 2 sensors-24-04293-t002:** Distribution of NSL-KDD dataset classes.

No.	Category	KDDTrain+	KDDTest+
1	Dos	45,927	7460
2	Probe	11,656	2421
3	R2L	995	2885
4	U2R	52	67
5	Normal	67,343	9711
Total		125,973	22,544

**Table 3 sensors-24-04293-t003:** Numerical values of categorical data.

Dataset	Feature Name	Numerical Value
UNSW-NB15	proto	0–132
	service	0–12
	state	0–7
NSL-KDD	protocol_type	0–2
	service	0–69
	flag	0–10

**Table 4 sensors-24-04293-t004:** Normalized dataset.

No.	Original Data		Normalized Data	
	**Dur**	**Sbytes**	**Dur**	**Sbytes**
1	0.000011	496	0.39285714	0.21634615
2	0.000008	1762	0.28571429	0.825
3	0.000005	1068	0.17857143	0.49134615
4	0.000006	900	0.21428571	0.41057692
5	0.00001	2126	0.35714286	1
6	0.000003	784	0.10714286	0.35480769
7	0.000006	1960	0.21428571	0.92019231
8	0.000028	1384	1	0.64326923
9	0	46	0	0
10	0	46	0	0

**Table 5 sensors-24-04293-t005:** Environment setup.

Hardware/Software	Specification/Version
OS	Windows 11 Enterprise 64-bit
CPU	12th Gen Intel(R) Core(TM) i7-12700H processor 2.30 GHz
Hard disk space	1.81 TB
GPU	NVIDIA RTX 3060
RAM	64.0 GB
Python	3.10.3
NumPy	1.26.3
TensorFlow	2.10.0
scikit-learn	1.4.0
Keras	2.10.0
Pandas	2.2.0
Matplotlib	3.8.2

**Table 6 sensors-24-04293-t006:** Confusion matrix.

		Predicted Class	
		**Attack**	**Normal**
Actual Class	Attack	True Positive (*TP*)	False Negative (*FN*)
	Normal	False Positive (*FP*)	True Negative (*TN*)

**Table 7 sensors-24-04293-t007:** Implementation specifications.

Hardware/Software	Specification/Version
Model	Raspberry Pi 3 Model B+
CPU	64-bit SoC @ 1.5 GHz
RAM	1 GB
Disk	32 GB
OS	Debian GNU/Linux, Version 11
Python	3.9.18
TensorFlow	2.12.0
Joblib	1.4.2

–The Raspberry Pi 3 Model B+ is a product of the Raspberry Pi Foundation. The organization is based in Cambridge, United Kingdom.–The SVM bagging model is exported as a pickle (.pkl) file and can be loaded onto a Raspberry Pi 3 Model B+ using Joblib. The LSTM stacking model is exported in the HDF5 (.h5) format and can be loaded onto a Raspberry Pi 3 Model B+ using TensorFlow.

**Table 8 sensors-24-04293-t008:** Distributions of classes between the original and balanced data for the UNSW-NB15 dataset.

Class	Original Dataset		Balanced Dataset	
	**Number of Rows**	**Percentage**	**Number of Rows**	**Percentage**
Normal	37,000	44.94	45,332	50
Attack	45,332	55.06	45,332	50
Total	82,332	100	90,664	100

**Table 9 sensors-24-04293-t009:** Distributions of classes between the original and balanced data for the NSL-KDD dataset.

Class	Original Dataset		Balanced Dataset	
	**Number of Rows**	**Percentage**	**Number of Rows**	**Percentage**
Normal	67,343	53.56	67,343	50
Attack	58,630	46.54	67,343	50
Total	125,973	100	134,686	100

**Table 10 sensors-24-04293-t010:** The number of selected features for each dataset.

Dataset	Feature Selection	Number of Features
UNSW-NB 15 dataset	Spearman Rank Correlation	16
	ANOVA	36
NSL-KDD dataset	Spearman Rank Correlation	15
	ANOVA	36

**Table 11 sensors-24-04293-t011:** Performance of Model 1.

Evaluation	UNSW-NB15 Dataset		NSL-KDD Dataset	
	**Validation**	**Training**	**Validation**	**Training**
no feature selection				
Accuracy (%)	80.68	84.17	43.08	53.51
Precision (%)	79.01	84.60	54.87	69.09
Recall (%)	82.95	83.47	50.00	50.06
F-measure (%)	79.53	83.79	30.12	34.99
Model size (Kb)	45,785		251,452	
Loading time (S)	3.3		17.1	
with Spearman rank correlation for feature selection				
Accuracy (%)	84.93	92.48	76.27	94.63
Precision (%)	83.36	92.67	78.10	94.68
Recall (%)	87.91	92.15	77.91	94.54
F-measure (%)	84.05	92.36	76.27	94.63
Model size (Kb)	28,095		14,785	
Loading time (S)	2.2		1.57	

**Table 12 sensors-24-04293-t012:** Model 2’s performance.

Evaluation	UNSW-NB15 Dataset		NSL-KDD Dataset	
	**Validation**	**Training**	**Validation**	**Training**
no feature selection				
Accuracy (%)	96.70	96.81	99.62	99.72
Precision (%)	96.74	96.86	99.62	99.72
Recall (%)	96.70	96.81	99.62	99.72
F-measure (%)	96.70	96.81	99.62	99.72
Model size (Kb)	1314		1308	
Loading time (S)	4.6		5.12	
with ANOVA for feature selection				
Accuracy (%)	96.59	96.92	99.73	99.77
Precision (%)	96.63	96.97	99.73	99.77
Recall (%)	96.59	96.92	99.73	99.77
F-measure (%)	96.59	96.92	99.73	99.77
Model size (Kb)	1278		1278	
Loading time (S)	4.5		4.7	

## Data Availability

Data are contained within the article.

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
