# Peer review of "Optimizing IoT Intrusion Detection Using Balanced Class Distribution, Feature Selection, and Ensemble Machine Learning Techniques"

_sensors, 2024, doi:10.3390/s24134293_

Round 1

Reviewer 1 Report

Comments and Suggestions for Authors

Comments to the authors:

This paper proposed a scheme to optimize IoT intrusion detection using a combination of balancing class and feature selection for preprocessing SMOTE technique to balance the rare classes of the dataset. The obtained results showed that the LSTM stacking with ANOVA feature selection model is a superior model to classify network attacks. This paper is solid in technique and contribution. This paper is also well-prepared.

Other comments:

1. References [4]-[5] are not cited in this paper. References in the reference list should be cited, follows the order given in the reference list.

2. The authors should add more introduction about the main difference between this paper and the previous works.

3. The motivation of this work is also presented more clearly.

4. The similarity score (27%) is high. The Reviewer think that this score should be below 20%.

5. There are several typos in this paper, which need to carefully corrected.  

Comments on the Quality of English Language

Minor editing of English language is required

Reviewer 2 Report

Comments and Suggestions for Authors

This study aims to optimize IoT intrusion detection using an integrated approach involving machine learning components such as feature selection, balanced class distribution, and ensemble machine learning. While the paper is well-written, it could be improved with the following points:

1. Class Balancing to Improve Performance: The proposed class balancing can create feature-focused models with high F1 scores for minority classes. However, this method may result in slightly lower overall accuracy due to reliance on proportional bias. Additionally, for small datasets, class balancing might not be effective as it could lead to feature loss and model prediction bias.

   The authors stated, "As future works, we would like to implement the proposed model on the Raspberry Pi. However, the device’s limited computational resources, including CPU, memory, and storage pose challenges. To address these constraints, we intend to explore model compression techniques that can reduce the model size without significantly compromising its performance. This optimization strategy is essential for achieving efficient computation times on the resource-constrained Raspberry Pi platform."

   Hence, dealing with small datasets and optimizing the model would be challenging.

2. Feature Selection and Class Correlation: The paper mentions that only highly correlated features with the class were considered to improve performance. It would be beneficial to discuss the impact on other classes and any potential loss or effect of this approach.

3. Performance Evaluation: The authors evaluated the performance of SVM bagging and LSTM stacking models with several parameters. However, the results section could be enhanced with more performance-related comparisons to substantiate the claimed optimization.

Comments on the Quality of English Language

English Language in this paper is fine. 

Reviewer 3 Report

Comments and Suggestions for Authors

Thank you for submitting your manuscript entitled "Optimizing IoT Intrusion Detection using Balanced Class Distribution, Feature Selection, and Ensemble Machine Learning Techniques". This work proposes a scheme to optimize IoT intrusion detection by using balancing class and feature selection for preprocessing. However, some areas in the manuscript require clarification and improvement. Therefore, I recommend major revisions with the following points addressed:

1. There are some issues with the citation of references. When citing references in your paper, it is crucial to include the citation the first time the relevant content appears. For example, when introducing some models in the Introduction, you need to add the corresponding references. For IoT intrusion detection, quantum communication [Science Advances 10, eadk3258 (2024)] and quantum machine learning [Research 6, 0134 (2023)] can be extremely helpful in defending against and detecting attacks, and it is suggested that the authors mention related work.

2. The content of the tables in the paper needs improvement. For example, in Tables 2 and 9, it is unnecessary to list specific category names, as it makes the tables lengthy and redundant. When comparing the performance of two models, it is recommended to bold key data in the tables to highlight important information. Additionally, consider including some graphs.

3. In the data encoding section, you have used target encoding. However, in many machine learning models, one-hot encoding is more commonly used. Target encoding introduces ordinal features, which might affect the final model results. Could the authors please explain this choice?

4. When introducing Spearman Rank Correlation, the paper mentions that if the correlation between two variables is close to 0, it indicates no significant correlation between them, and thus, both should be excluded. I am strongly skeptical about this. The lack of a significant correlation between two variables does not necessarily mean that they have no correlation with the target label.

5. There are some writing issues in the paper. For example, the F-measure formula on line 400 is incorrect. Additionally, the expression regarding the dataset on line 310 is not very clear.

I hope you find these comments useful, and I look forward to seeing the revised manuscript.

Comments on the Quality of English Language

Moderate editing of English language required

Reviewer 4 Report

Comments and Suggestions for Authors

1. Only one data set (UNSW-NB15) was used for verification in the experiment, and it is suggested to use more data sets for experiments to verify the generalization ability of the model.

2. SVM machine learning and LSTM deep learning models were used in the experiment, and no comparisons were made with other models such as Transformer and GRU.

3. In the introduction of experimental indicators in Section 4.1, accuracy, recall rate and accuracy are commonly used indicators of classification algorithms, which do not need to be introduced too much, and there is a problem of content redundancy.

4. The paper is not clear about why Spearman rank correlation and ANOVA are chosen instead of other feature selection techniques, and only two models are designed, one is Spearman rank correlation combined with SVM model, and the other is ANOVA combined with LSTM model. Please add experiments to show why these two combined methods are better than other methods.
